DATA RELEASE

# *De novo* transcriptome assembly and genome annotation of the fat-tailed dunnart (*Sminthopsis crassicaudata*)

Neke Ibeh[1,2,3,4,*], Charles Y. Feigin[1,5], Stephen R. Frankenberg[1],
Davis J. McCarthy[2,3], Andrew J. Pask[1] and Irene Gallego Romero[1,2,4,6,*]

1  School of BioSciences, The University of Melbourne, Parkville, VIC, Australia
2  Melbourne Integrative Genomics, The University of Melbourne, Parkville, VIC, Australia
3  Bioinformatics and Cellular Genomics, St Vincent's Institute of Medical Research, Fitzroy, VIC, Australia
4  Human Genomics and Evolution, St Vincent's Institute of Medical Research, Fitzroy, VIC, Australia
5  Department of Environment and Genetics, La Trobe University, Bundoora, VIC, Australia
6  Center for Genomics, Evolution and Medicine, Institute of Genomics, University of Tartu, Riia 23b, 51010, Tartu, Estonia

## ABSTRACT

Marsupials exhibit distinctive modes of reproduction and early development that set them apart from their eutherian counterparts and render them invaluable for comparative studies. However, marsupial genomic resources still lag far behind those of eutherian mammals. We present a series of novel genomic resources for the fat-tailed dunnart (*Sminthopsis crassicaudata*), a mouse-like marsupial that, due to its ease of husbandry and *ex-utero* development, is emerging as a laboratory model. We constructed a highly representative multi-tissue *de novo* transcriptome assembly of dunnart RNA-seq reads spanning 12 tissues. The transcriptome includes 2,093,982 assembled transcripts and has a mammalian transcriptome BUSCO completeness score of 93.3%, the highest amongst currently published marsupial transcriptomes. This global transcriptome, along with *ab initio* predictions, supported annotation of the existing dunnart genome, revealing 21,622 protein-coding genes. Altogether, these resources will enable wider use of the dunnart as a model marsupial and deepen our understanding of mammalian genome evolution.

**Submitted:** 04 December 2023

\* Corresponding authors. E-mail:
oibeh@student.unimelb.edu.au;
irene.gallego@svi.edu.au

Preprint submitted at https://doi.org/10.1101/2023.11.16.567318

**Subjects**  Genetics and Genomics, Bioinformatics, Evolutionary Biology

## DATA DESCRIPTION
### Background and context

Marsupials are a strikingly diverse mammalian group predominantly found in Australasia (Australia, Tasmania, New Guinea, and nearby islands), with several species also inhabiting the Americas [1, 2]. While many marsupials exhibit convergent traits with eutherian mammals [3–11], their adaptations to their respective niches encompass highly specialized physiology [12–18], behavior [19–21], and modes of reproduction [22–27], thereby representing a unique component of mammalian diversity. To date, marsupial studies have significantly contributed towards elucidating various aspects of mammalian biology, including reproductive physiology [24–27], sex determination [28–34], X-chromosome inactivation [24, 35–38], age-related obesity [39], postnatal development [15–18, 40–42], and

genome evolution [43–50], among others. Consequently, marsupials represent a critical comparative model system for advancing our understanding of mammalian biology.

Despite the importance of well-developed marsupial models, marsupial genomic resources still lag far behind those of their eutherian counterparts. Currently, there are 753 eutherian reference genome assemblies available through NCBI, but only 23 marsupial reference genomes (with just 9 RefSeq-annotated species). Some of these publicly available whole-genome assemblies include the gray short-tailed opossum (*Monodelphis domestica*) [51], the tammar wallaby (*Macropus eugenii*) [52], the Tasmanian devil (*Sarcophilus harrisii*) [43], the brown antechinus (*Antechinus stuartii*) [53], the koala (*Phascolarctos cinereus*) [54], the numbat (*Myrmecobius fasciatus*) [55], and the eastern quoll (*Dasyurus viverrinus*) [56], with genome assembly recovery of complete single-copy mammalian Benchmarking Universal Single-Copy Orthologs (BUSCOs) ranging from 73.1% to 92.4% [56]. The global transcriptomes generated for some of these species have BUSCO scores ranging from 76.4% to 84% [53, 55].

Recently, the fat-tailed dunnart (*Sminthopsis crassicaudata*, NCBI:txid9301) has emerged as a key laboratory marsupial model for understanding mammalian development and evolution [42, 57–61]. A nocturnal species belonging to the family Dasyuridae, the fat-tailed dunnart has adapted to a wide range of habitats and can be found across south and central mainland Australia [62] (Figure 1A and B). As one of the smallest carnivorous marsupials, adults weigh an average of 15 grams. Fat-tailed dunnarts exhibit some of the shortest known gestation times for mammals (13 days), with much of their development occurring postnatally. Fat-tailed dunnart neonates reside in their mother's pouch, thereby allowing continuous and non-invasive experimental access [63, 64]. The extremely altricial state of the dunnart young, along with very simple husbandry requirements, have facilitated the dunnart's role as a model species for comparative mammalian studies and conservation strategies.

However, the paucity of genomic resources for the fat-tailed dunnart has limited our understanding of this species at the gene level. As such, high-quality genome assembly and genome annotation have become increasingly important for investigations into the dunnart's unique biology. Recently, a draft fat-tailed dunnart genome assembly was released based on sequence data comprising ONT and PacBio long reads as well as Illumina HiSeq short reads [65]. While this scaffold-level assembly is a significant resource, an improved workflow was necessary in order to increase the genome's contiguity and completeness. Moreover, due to the absence of a *de novo* transcriptome, gene annotations had to be lifted over from the Tasmanian devil (*Sarcophilus harrisii*, GCF_902635505.1 - mSarHar1.11) to the dunnart scaffolds, thereby producing an incomplete representation of dunnart gene structure.

To address this knowledge gap, we present a comprehensive *de novo* transcriptome built from RNA-seq data from 24 samples, spanning 12 tissues. This global transcriptome has recovered 93.3% of complete mammalian BUSCOs, indicating its functional completeness. We also report the very first fat-tailed dunnart genome annotation. The genome annotation effort, made possible through the multi-tissue transcriptome assembly and *ab initio* predictions, yielded 21,622 protein-coding genes. Additionally, we provide an improved genome assembly that is 3.23 Gb in size with a scaffold N50 of 72.64 Mb. Annotated genomes and global transcriptomes are of paramount importance for attaching biological meaning to sequencing data. As such, this first-draft annotation and global transcriptome

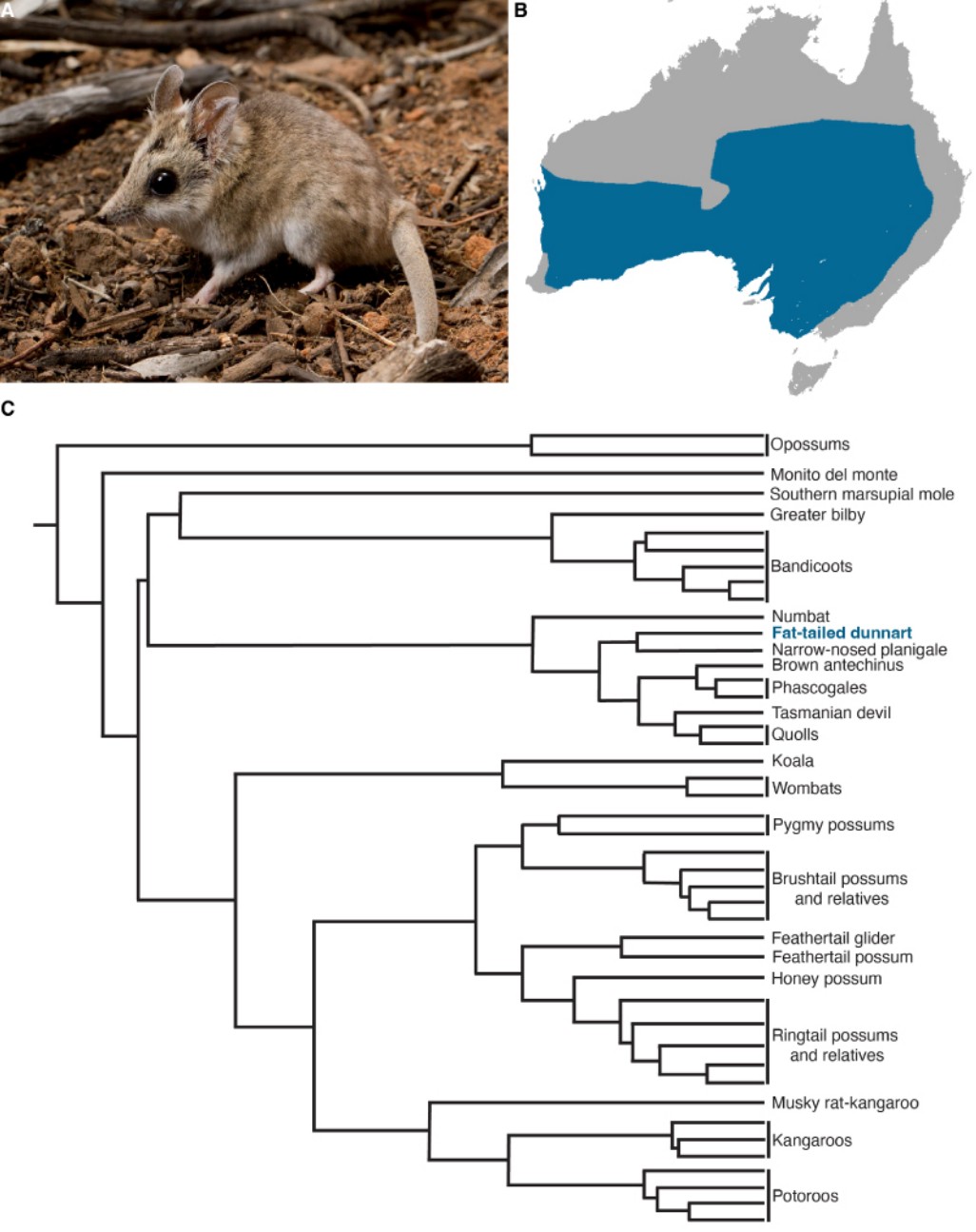

**Figure 1.   The fat-tailed dunnart (*Sminthopsis crassicaudata*).**
(A) Adult fat-tailed dunnart captured in Ned's Corner, Victoria (Photo credit: David Paul, Museums Victoria). (B) The fat-tailed dunnart's range across Australia (CC BY) [66]. (C) Phylogeny of extant marsupial orders (based on [67] and [68]). The fat-tailed dunnart (blue font) is a member of the order Dasyuromorphia.

can serve as tools with which the genomic architecture of the fat-tailed dunnart, an emerging marsupial model species, can be better understood. Most importantly, these comprehensive resources contribute to the growing body of research on marsupial genomics and are therefore invaluable tools for future mammalian studies.

## METHODS

### Draft genome assembly

Fat-tailed dunnart ONT (~18 Gb, including the new ONT library 20190606 in PRJNA1078592) and Pacific Biosciences CLR (~171 Gb) long reads [65], along with Illumina short reads (~447.5 Gb in 2 × 150 bp format) [50], were combined to produce an improved draft genome assembly. Briefly, *de novo* contigs were first assembled from long reads ≥10 kilobases using Flye v2.9 (RRID:SCR_017016) [69] (parameters: –pacbio-raw, –genome-size 3g –iterations 2 –scaffold). Uncollapsed haplotypes were removed using purge_dups [70] with automatic coverage threshold detection. A second round of scaffolding was then performed using LongStitch v1.0.1 [71] (mode: ntLink-arks with an estimated genome size of 3 Gb). The resulting assembly was then polished in two rounds using Pilon v1.24 (RRID:SCR_014731) [72] (parameters: –vcf –diploid –chunksize 10000000 –fix snps,indels,gaps –minqual 15). To do this, Illumina short reads were first filtered and trimmed with Trimmomatic v0.38 (RRID:SCR_011848) [73] (parameters: SLIDINGWINDOW:5:30, MINLEN:75, AVGQUAL:30). Reads were then aligned against the assembly using BWA-MEM2 [74] (parameter: –M), and the resulting alignments were filtered with Samtools view v1.11 (RRID:SCR_002105) [75] (parameters: –h –b –q 30 –F 3340 –f 3). All of the data used to assemble contigs came from females, with data from two individuals being combined (one female for the PacBio data and one female for the ONT data). Short reads were obtained from an individual of unknown sex and were thus excluded from the contig assembly stage. Benchmark mammal ortholog recovery for the assembly was determined using BUSCO v5.2.2 (RRID:SCR_015008) [76], in genome mode, using the Mammalia_odb v10 database of orthologs (9226 BUSCOs). BUSCO (v5.2.2) genome completeness scores were also computed for the numbat, koala, Tasmanian devil, Brown antechinus, tammar wallaby, gray short-tailed opossum, and the eastern quoll.

### Sample collection and sequencing

Adult and fetal fat-tailed dunnart tissues were collected for short-read Illumina sequencing from several individuals housed in a captive colony at the University of Melbourne. The tissues included late pregnancy allantois ($n$ = 3), amnion ($n$ = 3), distal yolk sac without vasculature ($n$ = 2), proximal yolk sac with vasculature ($n$ = 2), endometrium ($n$ = 4), ovary ($n$ = 3), oviduct ($n$ = 2), combined uterus and oviduct ($n$ = 1), testis ($n$ = 1), female liver ($n$ = 1), female eye ($n$ = 1), and prostate gland ($n$ = 1).

RNA samples were pooled in approximately equal proportions for Iso-Seq, namely, allantois, amnion, distal and proximal yolk sacs, endometrium, oviduct, ovary, testis, liver, eye, gastrula-stage conceptus, and late fetus. All RNA samples were extracted using Qiagen RNeasy Mini or Micro kits according to the manufacturer's instructions, with Illumina and Iso-Seq library construction and sequencing outsourced to Azenta Life Sciences (USA). For Illumina sequencing, this included rRNA depletion and strand-specific RNA library preparation, multiplexing, and sequencing on the NovaSeq platform, in a 2 × 150-bp (paired-end) configuration for 23 samples. Iso-Seq (poly-A selected and strand-specific) was performed using a PacBio Sequel II platform (1 sample, mean length of 5,400 bp). RNA Integrity Numbers (RIN) were generated using Bioanalyzer, and are available through Figshare [77].

### *De novo* transcriptome assembly

The raw RNA-seq reads were quality-checked using FastQC v0.11.9 (RRID:SCR_014583) [78]. Quality trimming of the short-read data was carried out using Trimmomatic v0.38 [73] (parameters: SLIDINGWINDOW:4:28, MINLEN:25, AVGQUAL:28). Post-trimming, 464M paired reads remained.

To generate a global dunnart transcriptome, the trimmed, paired-end RNA-seq reads were used as input to Trinity v2.13.2 (RRID:SCR_013048) [79]. We Applied default *in silico* read normalization and set the minimum assembled contig length to report to 200. Circular consensus reads were incorporated for Iso-seq long-read correction (parameter: –long_reads). Contig assembly was executed using three different k-mer settings: 25, 29, and 32. We chose these values because 25 and 32 are the minimum and maximum permitted values for the Trinity contig assembly step. Assembly statistics were obtained using the Trinity script TrinityStats.pl [79]. A reference-free evaluation of assembly quality was conducted using RSEM-EVAL, a component package of Detonate v1.11 [80]. RSEM-EVAL provides a weighted quality score using a probabilistic model. Although these scores are always negative, when comparing two assemblies, a higher value represents a higher-quality assembly. The completeness of the full-length assemblies was evaluated using Benchmarking Universal Single-Copy Orthologs (BUSCO) [76]. The BUSCO gene sets are comprised of nearly universally distributed single-copy orthologous genes representing various phylogenetic levels. Here, BUSCO v5.2.2 assessment was carried out in transcriptome mode using the Mammalia_odb v10 database of orthologs.

To quantify the RNA-seq read representation of the assembly, all reads were mapped back to the global transcriptome assembly using Bowtie2 v2.4.5 (RRID:SCR_005476) [81], setting a maximum of 20 distinct alignments for each read (parameter: –k 20). Transcript abundance was quantified using RSEM v1.3.3 [82], with Bowtie2 read alignments. Prior to annotation, transcript redundancy in the global transcriptome was reduced using CD-HIT v4.8.1 [83] with a homology threshold of 1 (parameter: –c 1) to avoid filtering out true isoforms.

### Transcriptome functional annotation

Functional annotation of the assembled transcripts was conducted using the Trinotate v3.2.2 [79] analysis protocol. First, Transdecoder v.5.5.0 [79] was used to identify all open reading frames (with a minimum length of 100 amino acids) and predict coding regions within transcripts. Sequence and domain homologies were captured by running BLAST+ v2.13.0 [84] (parameters: –max_target_seqs 1 –outfmt 6 –evalue 1e-5) against a combined protein database consisting of the UniProt/Swiss-Prot non-redundant protein sequences (RRID:SCR_002380) [85] from human (UP000005640), house mouse (UP000000589), Tasmanian devil (UP000007648), koala (UP000515140), tammar wallaby (txid9315), gray short-tailed opossum (UP000002280), and numbat (txid55782). Functional domains were identified by running a HMMer v3.3.2  [86] search against the PFAM v35.0 [87] database using the predicted protein sequences. Signal peptides and transmembrane domains were predicted using the SignalP v6.0 [88] and DeepTMHMM v1.0.24 [89] software tools, respectively.

### Genome annotation

Annotation of the dunnart draft genome was conducted using a combination of *ab initio* gene prediction algorithms and homology-based methods (Figure 2). First, genome repeats



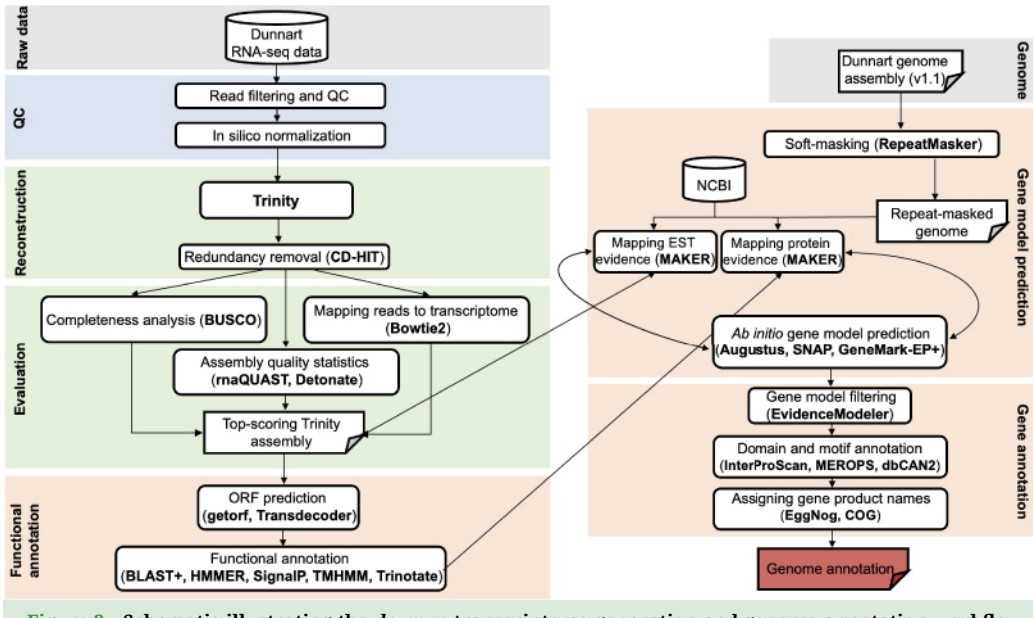

**Figure 2.** Schematic illustrating the *de novo* transcriptome generation and genome annotation workflow for the fat-tailed dunnart.

were masked using RepeatMasker v4.0.6 (RRID:SCR_012954) [90], with complex repeats being hard-masked while simple repeats were soft-masked. Preliminary gene models were constructed with MAKER2 (RRID:SCR_00530) [91] by aligning the assembled transcriptome and homologous protein sequences to the masked genome using minimap2 v2.26 [92] and DIAMOND v2.1.8 [93], respectively. Both cDNA (parameter: –model est2genome) and protein (parameter: –model protein2genome) alignments were polished with Exonerate v2.4.0 [94], producing high-quality alignments with precise intron/exon positions.

These preliminary gene models were then used to train the *ab initio* gene predictors SNAP (RRID:SCR_002127) [95], Augustus v3.4.0 (RRID:SCR_008417) [96], and GeneMark-EP+ v4.71 [97], all of which generated a statistical model representing the observed intron/exon structure in the genome. The gene model prediction process was iteratively run with MAKER2 (3 total rounds of prediction and re-training), thereby optimizing the performance of the *ab initio* gene predictors. For each round, prediction quality was evaluated using BUSCO scores. Consensus gene models were identified using EVidenceModeler v2.0.0 [98], with input weights set to 2 for high-quality *ab initio* predictions and to 1 for all other *ab initio* predictions and transcript/protein alignments. Gene models that lacked mRNA and protein homology support were excluded from the final annotation file. Lastly, gene names and putative protein functions were assigned using the aforementioned Trinotate output, as well as curated orthologous group and product names from InterProScan v5.60 (RRID:SCR_005829) [99], EggNOG v5.0 (RRID:SCR_002456) [100], MEROPS v12.4 [101], dbCAN3 v3.0.6 [102], and EuKaryotic Orthologous Groups (KOGs) [103].

## RESULTS

To generate a genome-level annotation for the fat-tailed dunnart, we began by producing an improved draft genome assembly. We employed a hybrid approach, which integrated the



**Table 1.** Fat-tailed dunnart genome assembly statistics compared to the numbat, koala, Tasmanian devil, brown antechinus, tammar wallaby, gray short-tailed opossum, and eastern quoll reference genomes currently available on NCBI.

| | Fat-tailed dunnart (this study) | Fat-tailed dunnart [65] | Numbat [55] | Koala [52] | Tasmanian devil [43] | Brown antechinus [53] | Tammar wallaby [52] | Gray short-tailed opossum [51] | Eastern quoll [56] |
|---|---|---|---|---|---|---|---|---|---|
| Genome size (Gb) | 3.23 | 2.84 | 3.42 | 3.19 | 3.17 | 3.31 | 3.07 | 3.59 | 3.14 |
| Number of scaffolds | 1,848 | 719 | 112,299 | - | 105 | 30,796 | 314 | 13 | 76 |
| Number of contigs | 2,569 | 1,154 | 219,447 | 1,906 | 444 | 106,199 | 829 | 2,268 | 507 |
| Scaffold N50 (Mb) | 72.64 | 28.02 | 0.22 | - | 611.3 | 72.7 | 489.7 | 538.3 | 628.5 |
| Scaffold L50 | 15 | 23 | 3,890 | - | 3 | 14 | 3 | 3 | 3 |
| Contig N50 (Mb) | 11.19 | 10.93 | 0.037 | 11.58 | 62.3 | 0.078 | 15.3 | 3.9 | 13.8 |
| Contig L50 | 81 | 78 | 24,796 | 85 | 14 | 12,151 | 60 | 244 | 72 |
| GC (%) | 36.18 | 36.25 | 36.30 | 39.05 | 36.04 | 36.20 | 38.80 | 38.00 | 36.19 |
| Complete Mammalian BUSCOs (v5.2.2, %) | 94.2 | 89.9 | 73.2 | 92.4 | 90.3 | 90.4 | 81.8 | 92.0 | 92.2 |

ONT and PacBio long-read data with Illumina paired-end short reads [50]. This resulted in a 3.23 Gb genome that contains 1,848 scaffolds and has a scaffold N50 of 72.64 Mb. The GC content of this draft genome is 36.2% (Table 1). The recovery of complete, single-copy mammalian BUSCOs was 94.2%. Together, these metrics are indicative of a high-quality genome assembly, with marked improvements over the existing dunnart draft genome and notably higher completeness and contiguity compared to other marsupial reference genomes currently available on NCBI (Table 1).

A *de novo* reconstruction of the dunnart transcriptome was conducted using a set of 24 RNA-seq samples originating from the liver, testis, prostate, ovary, oviduct, uterus, eye, whole neonate, allantois, amnion, distal yolk sac, proximal yolk sac, and endometrium. To ensure that the most representative assembly was obtained, we sought to identify the optimal k-mer length for the Trinity contig assembly step, considering k values of 25, 29, and 32 (Table 2). Given that reference-free transcriptome assembly relies on grouping overlapping sequences of read fragments of a predetermined size (i.e., the k-mer), identifying the optimal fragment size might yield a more accurate assembly. To assess this fragment size effect, we computed multiple assembly quality metrics, including the BUSCO completeness score (transcriptome mode) and the Detonate RSEM-EVAL score for each Trinity run. The RSEM-EVAL score represents the sum of three main factors: likelihood estimates of the read representation within the assembly, the assembly prior, which assumes that each contig is generated independently, and the BIC (Bayesian Information Criterion) penalty [80]. When comparing two assemblies, a higher RSEM-EVAL score is indicative of a more complete transcriptome assembly. In our comparison, the Trinity run with a k-mer setting of 29 produced the top-scoring assembly; thus, all subsequent analysis was carried out using this assembly.

This transcriptome assembly was composed of 2,093,982 assembled transcripts (including splicing isoforms), with a GC content of 40.2% and a mean transcript length of 830 bp (Table 2). The transcript N50 was 1,489 bp, and considering only the top 90% most highly expressed transcripts (a more accurate proxy for transcriptome quality [104]) produced an E90N50 of 3,430 bp. Sample reads that were mapped back to the assembly had a very high overall alignment rate (98%), with a high percentage mapped as proper pairs (94%). In addition, the global transcriptome had a 93.3% recovery of complete mammalian BUSCOs (Mammalia_odb v10 [76]). These values are in line with, or higher than, those reported from all other available marsupial transcriptome datasets (Table 3). Specifically,

**Table 2.** Summary of the *de novo* transcriptome assembly statistics for the Trinity k-mer optimization.

|  | Trinity-k25 | Trinity-k29 | Trinity-k32 |
|---|---|---|---|
| Total # of assembled Trinity transcripts | 2,588,090 | 2,093,982 | 1,960,023 |
| Mean transcript length (bp) | 731 | 830 | 922 |
| Transcript N50 (bp); E90N50 (bp) | 1,193; 2,990 | 1,489; 3,430 | 1,260; 3,154 |
| GC content (%) | 41.7 | 40.2 | 40.9 |
| Percentage of mapped RNA-seq PE reads (%) | 95 | 98 | 97 |
| Total BUSCO score (transcriptome mode) | C:92.1%, n:9226 | C:93.3%, n:9226 | C:93.0%, n:9226 |
| Detonate RSEM-EVAL score | $-6,136.0 \times 10^7$ | $-5,759.0 \times 10^7$ | $-6,110.0 \times 10^7$ |

**Table 3.** Summary of global transcriptomes from marsupial species.

|  | Numbat [55] | Tasmanian devil [105] | Brown antechinus [53] |
|---|---|---|---|
| Total # of assembled transcripts | 2,119,791 | 470,729 | 1,636,859 |
| Mean transcript length (bp) | 824 | – | 773 |
| Transcript N50 (bp) | 1,393 | 687 | 1,367 |
| Percentage of mapped RNA-seq PE reads (%) | – | 95 | 96 |
| Total BUSCO score (transcriptome mode) | 76.4% (v5.2.2) | – | 84% (v4.0.6) |

the global transcriptome assembly for the brown antechinus yielded 1,636,859 transcripts, with a mean length of 773 bp, a transcript N50 of 1,367 bp, a 96% alignment rate, and 84% complete BUSCOs [53]. The numbat global transcriptome contained 2,119,791 transcripts, a mean transcript length of 824 bp, a transcript N50 of 1,393 bp, and a BUSCO completeness score of 76.4% [55]. The Tasmanian devil transcriptome assembly consisted of 470,729 transcripts with an N50 of 687 bp and a 95% alignment rate of sample reads to the assembly [105].

Using a multi-pronged annotation approach of transcript and protein-level alignment, as well as *ab initio* gene prediction (Figure 2), we obtained 58,271 putative gene models for the fat-tailed dunnart draft genome (Table 4). Of these gene models, 21,622 were protein-coding (BLAST hits to UniProt/Swiss-Prot), which is in line with the reported gene numbers for the numbat (21,465) [55], the koala (27,669) [106], the Tasmanian devil (19,241) [43], the brown antechinus (25,111) [53], the tammar wallaby (15,290) [52], and the gray short-tailed opossum (21,384) [51] (Table 5). Furthermore, we predicted the putative function of the fat-tailed dunnart proteins using several curated protein databases (Table 4, Figure 3). We used InterProScan to identify conserved domains and assign Gene Ontology (GO) terms.

A total of 24,366 transcripts were assigned InterProScan terms, and 13,507 unique genes were assigned GO terms. The most common GO terms were intracellular anatomical structure (17,995 genes), organelle (17,140 genes), protein binding (17,071), cytoplasm (15,355 genes), and regulation of cellular processes (14,193 genes, Figure 3A). Notably, in another marsupial species, the woylie, cellular processes were also the most common GO term under the Biological Processes (BP) category [107]. Our GO annotations totaled 289,985, with a mean annotation level of 7.15 and a standard deviation of 2.7 (Figure 3B). Running an HMMer search against the PFAM database yielded 16,308 domains, while dbCAN3 and MEROPS analyses resulted in 212 and 1,053 predictions, respectively. Altogether, these results highlight valuable avenues through which we can deepen our understanding of marsupial biology at the gene level.



**Figure 3. Gene ontology (GO) analysis of the fat-tailed dunnart putative genes.**
(A) GO distribution by category (at level 3) for the fat-tailed dunnart gene set. The ontology categories are BP (Biological Process), MF (Molecular Function), and CC (Cellular Component). The top 20 terms are listed for each category. (B) Distribution of sequence annotations for each GO level.

**Table 4.** Fat-tailed dunnart gene and feature statistics.

|  | Fat-tailed dunnart |
|---|---|
| **General** | |
| Protein-coding genes | 21,622 |
| Predicted gene models | 58,271 |
| **Transcript level** | |
| mRNA | 50,091 |
| tRNA | 8,180 |
| Multiple exon transcripts | 44,109 |
| Single exon transcripts | 5,982 |
| Total exons | 246,391 |
| Average exon length | 146.1 |
| **Functional level** | |
| InterProScan terms | 24,366 |
| EggNOG terms | 29,372 |
| PFAM domains | 16,308 |
| dbCAN3 (CAZymes) | 212 |
| MEROPS (proteases) | 1,053 |
| GO | 13,507 |

**Table 5.** Fat-tailed dunnart gene counts compared to the numbat, koala, Tasmanian devil, brown antechinus, tammar wallaby, gray short-tailed opossum, and eastern quoll.

|  | Number of putative genes | Number of protein-coding genes |
|---|---|---|
| Fat-tailed dunnart (this study) | 58,271 | 21,622 |
| Numbat [55] | 77,806 | 21,465 |
| Koala [54] | 52,384 | 27,669 |
| Tasmanian devil [43] | 40,469 | 19,241 |
| Brown antechinus [53] | 55,827 | 25,111 |
| Tammar wallaby [52] | 122,304 | 15,290 |
| Gray short-tailed opossum [51] | 43,478 | 21,384 |
| Eastern quoll [56] | 29,622 | 14,293 |

## CONCLUSION

The increased availability of genomic resources for marsupial species is critical for fostering a deeper understanding of the evolutionary history of both eutherians and marsupials. In this study, we report an enhanced fat-tailed dunnart genome assembly measuring 3.23 Gb in length. The assembly is organized into 1,848 scaffolds, with a scaffold N50 value of 72.64 Mb. We generated a global *de novo* transcriptome assembly of the fat-tailed dunnart using RNA-seq short-read and long-read data, which were sampled from a diverse range of dunnart tissues. The transcriptome reconstruction consisted of 2,093,982 assembled transcripts, with a mean transcript length of 830 bp. The transcriptome BUSCO completeness score of 93.3% is the highest amongst all other published marsupial transcriptome BUSCOs (i.e., numbat and brown antechinus). The high overall alignment rate of reads from each of the tissues to the transcriptome (98%) further underscores that the *de novo* transcriptome is a highly accurate representation of the input reads. The dunnart draft genome annotation revealed 21,622 protein-coding genes, in line with previously reported marsupial gene counts. Overall, these resources provide novel insights into the unique genomic architecture of the fat-tailed dunnart and will therefore serve as valuable tools for future comparative mammalian studies.

## AVAILABILITY OF SOURCE CODE AND REQUIREMENTS

- Project name: Dunnart Genome Annotation
- Project home page: https://gitlab.svi.edu.au/igr-lab/dunnart_genome_annotation
- Operating system(s): Platform independent

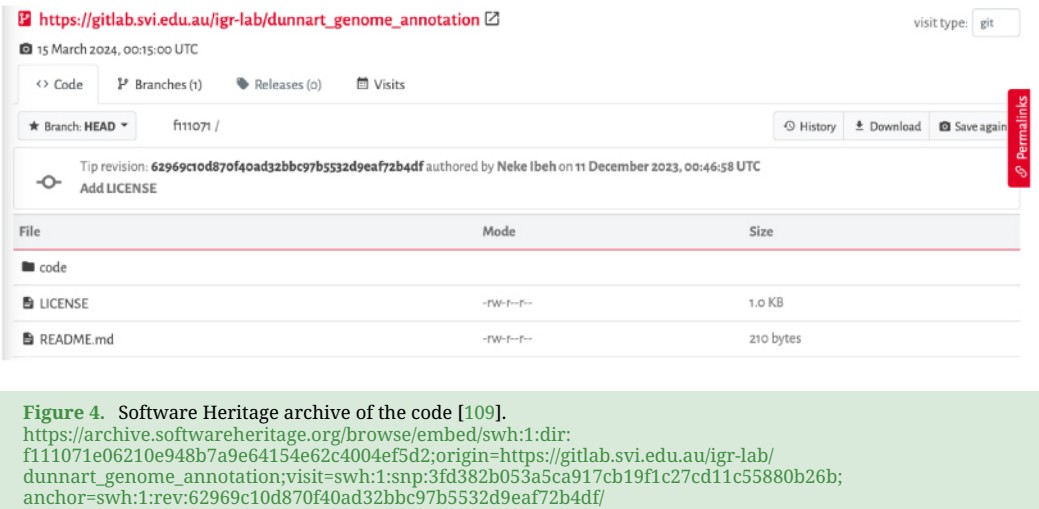

Figure 4. Software Heritage archive of the code [109].
https://archive.softwareheritage.org/browse/embed/swh:1:dir:
f111071e06210e948b7a9e64154e62c4004ef5d2;origin=https://gitlab.svi.edu.au/igr-lab/
dunnart_genome_annotation;visit=swh:1:snp:3fd382b053a5ca917cb19f1c27cd11c55880b26b;
anchor=swh:1:rev:62969c10d870f40ad32bbc97b5532d9eaf72b4df/

- Programming language: Shell, Python, Perl
- License: MIT.

## DATA AVAILABILITY

The fat-tailed dunnart transcriptome, draft genome, and genome annotation are available through Figshare [108]. The scripts for reproducing the genome annotation workflow have been made available in gitlab and archived in Software Heritage (Figure 4) [109]. All raw sequencing reads have been deposited at the National Center for Biotechnology Information (NCBI) Sequence Read Archive under the accession numbers PRJNA1078592 (genomic reads) and PRJNA1028148 (RNA-Seq).

## LIST OF ABBREVIATIONS

BLAST: Basic Local Alignment Search Tool; bp: base pair; BUSCO: Benchmarking Universal Single-Copy Orthologs; CDS: coding sequences; Gb: Gigabase; Kb: Kilobase; Mb: Megabase; NCBI: National Center for Biotechnology Information; ONT: Oxford Nanopore Technologies; PacBio: Pacific Biosciences; PE: paired-end; RNA-seq: RNA sequencing.

## DECLARATIONS

### Ethics statement

All sample collection was approved by the University of Melbourne Animal Ethics Committee (Project ID 10206).

### Competing interests

The authors declare that they have no competing interests.

### Authors' contributions

SF, AJP, CYF, and IGR conceived the project. SF collected and prepared the samples. NI assembled and annotated the global transcriptome. CYF assembled the draft genome. NI annotated the draft genome. NI drafted the manuscript with input from all authors. All authors read and approved the final version of the manuscript.

## Funding

This work was supported by the Australian Research Council Discovery Project DP210102645 to AJP. IGR was partially supported by the European Union through the Horizon 2020 Research and Innovation Program under Grant No. 810645 and the European Union through the European Regional Development Fund Project No. MOBEC008.

## Acknowledgements

NI was supported by the University of Melbourne Research Training Program Scholarship, the Rowden White Scholarship, the Dame Margaret Blackwood Soroptimist Scholarship, and the St. Vincent's Institute Top-up Scholarship. St. Vincent's Institute acknowledges the infrastructure support it receives from the National Health and Medical Research Council Independent Research Institutes Infrastructure Support Program and from the Victorian Government through its Operational Infrastructure Support Program.

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
