## [Editor Report]

Editor’s AssessmentMarsupial species are invaluable for comparative studies due to their distinctive modes of reproduction and development, but there are a shortage of genomic resources to do these types of analyses. To help address that data gap multi-tissue transcriptomes and transcriptome assemblies have been sequenced and shared for the fat-tailed dunnart (Sminthopsis crassicaudata), a mouse-like marsupial that due to is ease of breeding is emerging as a useful lab model. Using ONT nanopore and Pacbio long-reads and illumina short reads 2,093,982 transcripts were sequenced and assembled, and functional annotation of the assembled transcripts was also carried out. Some addition work was required to provide more details on the QC metrics and access to the data but this was resolved during review. This work ultimately producing dunnart genome assembly measuring 3.23 Gb in length and organized into 1,848 scaffolds, with a scaffold N50 value of 72.64 Mb. These openly available resources hopefully provide novel insights into the unique genomic architecture of this unusual species and provide valuable tools for future comparative mammalian studies.

---

## [Reviewer Report]

Reviewer name and names of any other individual's who aided in reviewer Qiye LiDo you understand and agree to our policy of having open and named reviews, and having your review included with the published papers. (If no, please inform the editor that you cannot review this manuscript.)YesIs the language of sufficient quality?YesPlease add additional comments on language quality to clarify if needed
Are all data available and do they match the descriptions in the paper? YesAdditional CommentsAre the data and metadata consistent with relevant minimum information or reporting standards? See GigaDB checklists for examples <a href="http://gigadb.org/site/guide" target="_blank">http://gigadb.org/site/guide</a>YesAdditional CommentsIs the data acquisition clear, complete and methodologically sound?YesAdditional CommentsIs there sufficient detail in the methods and data-processing steps to allow reproduction?YesAdditional CommentsIs there sufficient data validation and statistical analyses of data quality? YesAdditional CommentsIs the validation suitable for this type of data?YesAdditional CommentsIs there sufficient information for others to reuse this dataset or integrate it with other data?YesAdditional CommentsAny Additional Overall Comments to the AuthorFor the ONT, PacBio and Illumina data for genome assembly, is there any new data that was generated in this manuscript? Are all of the data collected from the same individual? If so, what is the gender of the individual for genome assembly? It will be appreciated to make this information clear to readers.   Page 3: I think "Pacific Biosciences CRL" should be modified to "Pacific Biosciences CLR".RecommendationAccept

---

## [Reviewer Report]

Upload additional filesDRR-202312-01/form/gx-DR-1701669977_EP (2).pdfReviewer name and names of any other individual's who aided in reviewer Emma PeelDo you understand and agree to our policy of having open and named reviews, and having your review included with the published papers. (If no, please inform the editor that you cannot review this manuscript.)YesIs the language of sufficient quality?YesPlease add additional comments on language quality to clarify if needed
Are all data available and do they match the descriptions in the paper? NoAdditional CommentsThe figshare link doesn't work, but I'm presuming this is because the paper hasn't been published? Will data be accessioned in the GigaScience Database to ensure accessiblity?   The illumina short-read genomic and RNAseq datasets are available through NCBI and match descriptions in the paper.   I was unable to find the raw PB and ONT data from [68] that was used to generate the genome assembly. The authors of [68] indicate these datasets are available in supplementary table 3, but if you click through the figshare link in this table the raw data isn't there, nor anywhere else listed in the data availability section. Can the authors please clarify the location of the raw data and update the data availability section of this manuscript accordingly. Are the data and metadata consistent with relevant minimum information or reporting standards? See GigaDB checklists for examples <a href="http://gigadb.org/site/guide" target="_blank">http://gigadb.org/site/guide</a>YesAdditional CommentsAccess to the GigaDB accession hasn't been provided, so I am unable to determine if the data and metadata is consistent with minimum information reporting standards according to the GigaDB checklists. Is the data acquisition clear, complete and methodologically sound?YesAdditional CommentsSome minor clarifications are required, see comments in the PDF. For example, please include detail on how RNA quality was determined (e.g. RIN numbers) and provide more detail regarding method of library preparation, flowcell and instrument used for Illumina sequencing. Is there sufficient detail in the methods and data-processing steps to allow reproduction?YesAdditional CommentsThe only detail lacking is the method of transcript quantification used to determine the top 90% most highly expressed transcripts. Is there sufficient data validation and statistical analyses of data quality? YesAdditional CommentsIs the validation suitable for this type of data?YesAdditional CommentsData validation is suitable, however I would like to see a comparison of v1.1 genome assembly with other marsupial genome assemblies. Is there sufficient information for others to reuse this dataset or integrate it with other data?YesAdditional CommentsAny Additional Overall Comments to the AuthorThis study is an important addition to marsupial omics resources, and I was excited to see such a comprehensive set of transcriptomes. My main comment is the need to explain and discuss the initial assembly (v1) in the introduction to provide context for the improved assembly. See comments in the attached PDF. RecommendationMinor Revision